# Synthesis of Nano-Magnetite from Industrial Mill Chips for the Application of Boron Removal: Characterization and Adsorption Efficacy

**DOI:** 10.3390/ijerph18041400

**Published:** 2021-02-03

**Authors:** Mohammed Umar Abba, Hasfalina Che Man, Raba’ah Syahidah Azis, Aida Isma Idris, Muhammad Hazwan Hamzah, Mohammed Abdulsalam

**Affiliations:** 1Department of Biological and Agricultural Engineering, Faculty of Engineering, Universiti Putra Malaysia, Serdang 43400, Selangor, Malaysia; gs51611@student.upm.edu.my (M.U.A.); hazwanhamzah@upm.edu.my (M.H.H.); 2Department of Agricultural and Bioenvironmental Engineering, Federal Polytechnic Mubi, Mubi 650221, Nigeria; 3Smart Farming Technology Research Centre, Level 6, Block Menara, Faculty of Engineering, Universiti Putra Malaysia, Serdang 43400, Selangor, Malaysia; 4Department of Physics, Faculty of Science, Selangor Universiti Putra Malaysia, Serdang 43400, Selangor, Malaysia; rabaah@upm.edu.my; 5Materials Synthesis and Characterization Laboratory (MSCL), Institute of Advanced Technology (ITMA), Universiti Putra Malaysia, Serdang 43400, Selangor, Malaysia; 6Department of Chemical Engineering, Faculty of Engineering, Segi Universiti Malaysia, Petaling Jaya, Serdang 43400, Selangor, Malaysia; aidaisma@segi.edu.my; 7Department of Agricultural and Bio-Resources, Ahmadu Bello University, Zaria 810107, Nigeria; m.abdul_22@yahoo.com

**Keywords:** nano-magnetite, boron, adsorption, regeneration, isotherm, kinetics

## Abstract

The present study synthesized nano-magnetite (Fe_3_O_4_) from milled steel chips using the high energy ball milling (HEBM) method, characterized it, and then utilized it as a sorbent to remediate boron concentration at various pH (4–9), dosages (0.1–0.5 g), contact times (20–240 min), and initial concentrations (10–100 mg/L). The nano-sorbents were characterized based on SEM structure, elemental composition (EDX), surface area analysis (BET), crystallinity (XRD), and functional group analysis (FTIR). The highest adsorption capacity of 8.44 mg/g with removal efficiency of 84% was attained at pH 8, 0.5 g dosage, contact time of 180 min, and 50 mg/L initial concentration. The experimental data fit best with the pseudo-second-order kinetic model with R^2^ of 0.998, while the Freundlich adsorption isotherm describes the adsorption process with an R^2^ value of 0.9464. A regeneration efficiency of 47% was attained even after five cycles of reusability studies. This efficiency implies that the nano-magnetite has the potential for sustainable industrial application.

## 1. Introduction

The industrial development and geometric growth in global populations is accompanied by the excessive generation of boron from industrial and municipal waste [1]. The U.S. Geological Survey estimated in 2006 that 4300 metric tons of boron were generated in 2006 and the figure rose geometrically to 9400 metric tons in 2016 owing to heavy industrial development and rapid population rise [2]. Boron is an essential trace element for animals and plants as a micronutrient which plays a prominent role in plant growth and development. However, an excessive quantity is toxic or even harmful to both plants and animals [3]. Additionally, higher concentration of boron in surface waters can be triggered by geochemical background and anthropogenic factors. Many studies have indicated that natural (weathering of rocks) and industrial wastewater discharge are the primary sources of the releasing of boron into the environment [4,5].

Several reports have stated that boron at an elevated concentration is a harmful chemical which is present in water and wastewater [6,7]. A high proportion of boron can provoke toxic effects on the environment and human health such as acute vomiting, nausea, diarrhea, dermatitis, and lethargy [8,9]. Furthermore, boron presents a clear effect on plants, including meristematic growth in tissues, disruption of leave and root growth, thickening of leaves, and disruption of cell development together with a delay in enzyme reactions [10]. Other effects of boron may include yellowish dots on fruits and leaves along with swift deterioration of plants when it occurs in high proportions [11]. Hence, the need to regulate the concentration of boron in water and wastewater through water treatment processes prior to final release into the environment is vital. Furthermore, the World Health Organization (WHO) recommends 2.4 mg/L maximum boron concentration in drinking water [12], whereas the concentration of boron in irrigation water is set at 1 mg/L or less [13]. Thus, it is a concern for both human health and agricultural reasons. It is a fact that the need for fresh water throughout the world is increasing, with a necessity to implement innovative methodologies.

Numerous techniques for boron removal from wastewater have been examined and these include precipitation-coagulation [14], membrane filtration [15], and adsorption [16]. However, the removal of boron using the aforementioned techniques still presents a huge challenge due to high chemical demand, operational cost, and physicochemical characteristics of boron such as lower molecular size, being nonionic at neutral pH, and its rapid rate of diffusion [17]. Therefore, sustainable, cost-effective, waste-driven, efficient, and potent adsorbents may be applied in the removal of boron in an adsorption process. This technique not only enhances waste management and recovery drive, but also advances the application of low-cost and eco-friendly adsorbents. In this context, adsorption is regarded as one of the most promising technologies for efficient boron removal due to its simplicity in design, facile application, affordability, and its non-toxic nature [18]. The adsorbent required for an adsorption process may be produced from different sources such as cellulose [19], soil [20], fly ash and coal [21], activated sludge [4], boron-selective resins [22], and coconut shell [23], among others. However, the adsorptive capacity relies on the intrinsic properties of the precursor, the carbonation-activation method, and the behavioral properties of the targeted adsorbate [24]. For example, Kluczka et al. [25] conducted an adsorption study using gly-resin to eliminate boron from wastewater and recorded an optimum adsorption capacity of 1.6 mg/g, at pH 9. Similarly, Kluczka et al. [26,27] applied cobalt (II) doped as a remediating sorbent to remove boron from wastewater at a pH of 8.5. The optimum adsorption capacity of 2.5 mg/g was attained. Man et al. [28] reported an adsorptive capacity of 4.23 mg/g when unmodified rice husk was used as a precursor at an alkaline pH. Irrespective of the adsorptive capacity, the adsorption performance has a strong correlation with pH [29]. However, studies on the removal of boron from water using in-house synthesized nano-magnetites from mill scale waste as a precursor are very scarce and need to be explored. 

Generally, nanomaterial-based adsorbents enhance the adsorption capacity and the efficiency of the adsorption process [30]. In particular, the nano-magnetite (Fe_3_O_4_) precursor exhibits multifunctional properties including superior superparamagnetism, characterized clusters of microspores, wider surface area, and non-toxicity [31,32]. These unique properties make it a suitable and robust precursor for the adsorptive removal of boron [33]. Besides, it does not generate secondary waste (sludge) and can be easily recycled and reused on an industrial scale [31,32,34]. 

In this regard, the present study mainly focuses on synthesizing nano-magnetite (Fe_3_O_4_) from locally available industrial milled chips using a high-energy ball milling technique. The resultant nano-magnetite (Fe_3_O_4_) sorbent was then characterized using SEM (scanning electron microscopy), EDX (Energy dispersive x-ray spectroscopy), and FTIR (Fourier-transform infrared spectroscopy) analysis. The boron adsorption performance was examined at a varied pH (4–9), contact time (20–240 min), dosage (0.1–0.5 g), and initial concentration (10–100 mg/L). Regeneration studies were conducted by repeatedly applying the used nano-magnetite (Fe_3_O_4_) after desorption processes to examine its reusability. The experimental data were evaluated with kinetics and isotherm models to further determine the adsorption process of the synthesized nano-magnetite (Fe_3_O_4_). 

## 2. Materials and Methods 

### 2.1. Materials and Chemicals

Boric acid, H_3_BO_3_, was procured from Aldrich (Chemical Industry Stock Co., Ltd. Yanwumiao Linhua Ind Park Dangyang City Yichang 444100, China,) for the preparation of a boron standard solution (1000 ppm). A pH 5-SS (spear pH Tester) and HY-8 variable speed reciprocal vibrator were used in the experiment for the pH measurement and agitation, respectively. A UV spectrophotometer (Hach DR4000u) was used to analyze the boron concentration at 605 nm wavelength. The initial boron concentration was determined using the standard procedure (APHA, 2005) [35] and the remaining bulk of the sample was preserved in a chiller at 4 °C. The milled scale chips were obtained from a steel company in Malaysia. Distilled water used in the batch experiments was acquired from the refinement system Milli-Q water (18 MO cm).

### 2.2. Synthesis of Nano-Magnetite (Fe_3_O_4_) from Industrial Milled Chips

Figure 1 depicts the procedure employed for the synthesis of nano-magnetite (Fe_3_O_4_) from milled chips. Initially, the milled chips were cleaned extensively using distilled water and dried at 104 °C for 24 h, then smashed into micron-size using a conventional milling machine. This process was carried out steadily for 48 h and the resultant micro-sized magnetite (Fe_3_O_4_) was further purified using magnetic separation technique (MST). The MST promotes the separation of magnetic and non-magnetic particles. Subsequently, the purified micron magnetite was oven-dried overnight at 104 °C and then transferred into an airtight container. Furthermore, the strong magnetic particles were separated from the weak ones using curie temperature separation technique (CTST), as shown in Figure 1. This technique is in accordance with previously adopted procedures [36,37]. The separated strong magnetic particles were later oven-dried for 24 h and then exposed to high-energy ball milling (HEBM) for 3 h to obtain a nano-sized nano-magnetite [38]. 

### 2.3. Characterization of Synthesised Nano-Magnetite 

An automated device (Micrometrics ASAP2020) was used to evaluate the specific surface area for the adsorption and desorption of N_2_ at 77 K. The specimen was outgased at 200 °C for 4 h in light inert gas pressure prior to their use. The structural morphology of the synthesized adsorbent was analyzed with SEM/EDX with Hitachi Co., Japan Model No. S3400N. The FTIR provides information on functional groups existing in the synthesized adsorbent. The Bruker-Tensor 27 IR appliance with standard KBr-pellet method in the spectral range 400–4000 cm^−1^ with 2 cm^−1^ resolution identified FTIR spectra of the nano-magnetite adsorbent. The X-ray diffraction (XRD) method was used to analyze the chemical composition of the synthesized nano-magnetite adsorbent at Cu Kα radiation (2θ spectrum = 20°–80°; phase = 0.05° 2θ; time per step = 0.2 s) using an X’Pert Philips PW3040 diffractometer.

### 2.4. Batch Adsorption Study 

In the batch adsorption experiment, 6.0 g of boric acid (H_3_BO_3_) was dissolved into 1.0 L of distilled water during the preparation of boron standard solution (1000 ppm). A conical flask of 250 mL was filled with 100 mL of boric acid at various initial concentrations (10–100 mg/L), pH range (4–9), and adsorbent dosages (0.1–0.5 g). The sample was positioned on an orbital shaker at 150 rpm at contact time 20–240 min. The boron concentrations in the sample were determined using a UV-Vis spectrophotometer (Hach DR/4000u). 

### 2.5. Adsorption Isotherm

The Freundlich and Langmuir adsorption isotherm models were utilized to define the dispersion of the adsorbent, building upon certain hypothesis regarding the heterogeneity and homogeneity of the adsorbent. In this regard, they indicate the proportion of the liquid solute (adsorbate) deposited on the surface solid phase (adsorbent) and the remnant in solution at a given time and concentration. Furthermore, the equilibrium isotherm was plotted by the values of the fluid phase concentration against solid phase concentration [39].

#### 2.5.1. Langmuir Isotherm

The Langmuir adsorption describes gas–solid phase adsorption and measures and contrasts the adsorption capacity of different adsorbents [40]. Equally, the Langmuir isotherm was used in balancing (dynamic equilibrium) the relative rate of adsorption and desorption for surface coverage. Desorption proportionality is when the adsorbent surface is covered, whereas adsorption is proportional to the adsorbent surface area that is open [41]. Equation (1) expresses the linearized form of the Langmuir equation [42].
(1)$$\frac{Ce}{qe}=\frac{1}{\mathrm{qm}\;(K_{L)}}+\frac{Ce}{\mathrm{qm}\;}$$
where ***C_e_*** is concentration of adsorbate at equilibrium (mg/g). ***K_L_*** is Langmuir constant associated with adsorption capacity (mg/g). The slope, 1/q_m_ and intercept, 1/***K_L_***. q_m_ could be determined from a plot of *C_e_*/*q_e_* against *C_e_* correlation coefficient R^2^. 

#### 2.5.2. Freundlich Isotherm 

The Freundlich isotherm describes the adsorption process on a heterogenous surface [43]. It describes the heterogeneity of the surface and the exponential distribution of active sites and their energy [44]. Equation (2) expresses the linearized model of the Freundlich isotherm [45].
(2)$${\log\;q}_{e}=\log K_{f}+\frac{1}{n}{\mathrm{logC}}_{e}$$

*K_f_*, n and r^2^ can be determined from the slope, intercept and correlation coefficient by plotting lnq_e_ against lnC_e_ [46]. The applicability of an isotherm can be determined through the coefficient of determination (R^2^) [47].

#### 2.5.3. Re-Usability 

The nano-magnetite was recovered by the solvent desorption technique when the active pore sites reached equilibrium. The nano-magnetite was removed from the solution by an external magnet, followed by dipping into HCL solution and mixing for 180 min at 26 °C. The resultant nano-magnetite was then washed with distilled water to attain neutral pH and then exposed for 1 h at 60 °C. The regenerated nano-magnetite was then re-applied in line with previous studies [48,49]. The re-usability efficiency (RE%) was determined using Equation (3):(3)$$RE=\frac{q_{reg}}{q_{ori}}\times 100\%$$
where *q_reg_* and *q_org_* are their respective adsorption capacities per unit mass of the regenerated and original adsorbents.

### 2.6. Analytical Method

The boron concentration was determined by carmine method 8015 at 605 nm wavelength [50]. The procedure involves adding boron ver 3 reagents into a 75 mL concentrated H_2_SO_4_ solvent. After 5 min reaction, 2 mL of specimen and deionized water was accurately pipetted into 125 mL flasks. In each flask, 35 mL of boron ver 3/H_2_SO_4_ solution was added. The prepared sample was inserted into the cell holder of the UV-Vis spectrophotometer (DR/4000u) at the wavelength of 620 nm. The adsorption capacity q_e_ (mg/g) and removal efficiency Re (%) of boron were determined from Equations (4) and (5), respectively [51,52].
(4)$$q_{e}=\frac{\left(C_{o}-C_{F}\right)V}{M}$$
where $q_{e}$ is the equilibrium boron concentration (mg/g), Co and C_F_ are the initial and equilibrium concentrations of boron in the sample (mg/L), respectively, M is the adsorbent mass (g), and V (L) is the volume of the boron solution. Boron removal efficiency was determined from Equation (5) [53].
(5)$$Re\;\left(\%\right)=\;\frac{C_{o}-C_{F}}{C_{o}}\;\times 100$$

## 3. Results and Discussion

### 3.1. Characterization of Synthesized Nano-Magnetite

#### 3.1.1. Functional Group Analysis

The vibration and stretching of functional groups of nano-magnetite particles before and after adsorption are shown on the FTIR spectra (Figure 2). The functional groups associated with the adsorption of boron by nano-magnetite are Fe-O and OH^−^ groups. This phenomenon is in agreement with previous studies [54,55,56]. The stretching of O-H over the surface of the nano-magnetite [57] at 3463 cm^−1^ moved to 3469 cm^−1^ after adsorption. This might be due to the hydrogen bond formed during the adsorption process between magnetite and boric acid [56]. The B-O stretching on the magnetite affirms that boron was adequately adsorbed on the magnetite [58]. The vibration peaks ascribed to Fe-O move from 649 cm^−1^ to 643 cm^−1^ after adsorption [33,59], which might be due to the new B atom joined to the bond between the adsorbate and Fe-O [33]. The peak at 1739 cm^−1^ showed that some water-binding molecules could occur on the surface of the nano-magnetite after boron adsorption [59]. After adsorption, the stretching of O-H on the nano-magnetite surface at 3463 cm^−1^ moved to 3469 cm^−1^ [57], possibly caused by the hydrogen bond during the adsorption of boric acid on the nano-magnetite. The change in wave number of O-H, B-O, and Fe-O bonds after adsorption showed that adsorption of boron on magnetite has taken place via development of new bond, Fe-O-B [60].

#### 3.1.2. Morphological Analysis of Synthesized Nano-Magnetite

The SEM characterization of the magnetite (before and after adsorption) was conducted to view the morphological structure as well as the particle size and size distribution of the nano-magnetite, as presented in Figure 3a,c respectively. Figure 3a exhibits a rough and denser surface. The adsorption of boron resulted in multiple attachments on the rough surface and a cloudier structure as depicted in Figure 3c. It can be observed that there is a change in the uniform structure, which might be attributed to the adsorption of boron ions on the nano-magnetite sorbent. The nano-magnetite sorbent exhibits magnetic behavior, creating more negative charges, which are automatically attracted to the positively charged ion in the boron solution. A similar surface structure was exhibited in a prior study [54]. The average particle size of Fe_3_O_4_ before adsorption as presented in Figure 3b was estimated to be 87.41 nm. However, after adsorption of boron, the average particle size increased to 200.88 nm as depicted in Figure 3d.

#### 3.1.3. Elemental Analysis of Synthesized Magnetite

The elemental composition of the nano-magnetite is shown in Figure 4. Both Fe and O appeared on the EDX spectra before the adsorption process with the absence of boron (Figure 4a) whereas (Figure 4b) confirms the presence of B, Fe, and O. The presence of boron on the spectra indicates the adsorption of boron by magnetite. A similar trend was observed when Lakshimi et al. (2016) utilized iron oxide nanoparticles for the adsorptive removal of heavy metals [61].

#### 3.1.4. Assessment of Specific Surface Area of Synthesized Nano-Magnetite

The plots of the nitrogen adsorption isotherm and the pore size distribution of magnetite are presented in Figure 5. The shapes of the isotherms are identical and mean that they are type II1 according to the classification of the International union of pure and applied chemist (IUPAC) adsorption isotherm [62]. The quantity of nitrogen adsorbed increased slightly at relative pressure <0.9 whereas the quantity adsorbed rapidly increased at relative pressure >0.9.

The Brunauer-Emmett-Teller (BET) surface area and average pore size of the synthesized nano-magnetite are 5.8729 m^2^/g and 5.6637 nm, respectively. After adsorption, the specific surface area of the adsorbent decreased to 3.92 m^2^/g while the pore size also reduced to 5.2092 nm. The adsorption capacity of nano-magnetite relies on the availability of pore sites and the time taken for the adsorption process. The reduction in the pore size after adsorption could be attributed to blockage of the pores due to adsorption. Furthermore, the reduction in pore size after adsorption may be attributed to the blockage of the pores due to adsorption. A similar finding was reported by Ref. [50]. 

#### 3.1.5. X-ray Diffraction Pattern

Figure 6 shows the XRD spectra of the nano-magnetite. As seen from the spectra, all the diffraction peaks are assigned to the magnetic cubic structure with reference code: ICSD 98-010-9823, crystal arrangement: cubic; space group: F d 3 m; lattice parameter: a = b = c: 8.3860. The diffraction peaks at 2θ values 30.1°, 37.7°, 43.2°, 45.0°, 58.2°, 63.8°, and 72.4° are compatible with hkl plane of 022, 113, 004, 131, 115, 044, and 026, respectively. There are no other peaks related to another material detected from the XRD result, which affirmed that the nano-magnetite is pure magnetite (Fe_3_O_4_).

### 3.2. Batch Adsorption Analyses

#### 3.2.1. Contact Time Dependency

An excellent adsorbent should possess a short period to reach adsorption equilibrium so that it can be applied efficiently to remove contaminants in a minimum time [63]. The boron removal efficiency increased rapidly from the early 20 min, followed by slower and steadier removal after 3 h (Figure 7). More than 50% boron removal was achieved within 20 min, which could be attributed to the unsaturated surface of the nano-magnetite. This trend agrees with similar studies reported earlier [3,54]. A similar pattern was also exhibited when researchers [64] studied the effect of pH and contact time on the removal of Hg(II). The boron adsorption capacity is presented in Figure 8. The maximum adsorption (8.4 mg/g) was attained after 180 min.

#### 3.2.2. Initial Concentration Dependency

The initial concentration provides the motivating force needed to conquer the mass transfer wall between the adsorbent and adsorbate media [65]. Hence, a higher initial concentration may increase the adsorption capacity. Noticeably, boron removal is lower at a low concentration due to fewer adsorbates in the solution to occupy active sites on the adsorbent and the quantity of boron adsorbed increases with the upsurge in boron concentration. The boron concentration in the stock solution varied between 10–100 mg/L with a constant dose of 0.5 g of nano-magnetite at pH 8. Hence, an increase in the concentration of boron corresponded to an increase in magnetite removal as shown in Figure 9. When equilibrium is reached, the adsorbent becomes saturated. This finding is consistent with a previous study [66].

#### 3.2.3. Dosage Dependency

The adsorbent dose plays an essential role in the adsorption process, as it governs the capability of the adsorbent for a given solution. The higher the dosage, the more available sites for sorption to take place [18]. As shown in Figure 10, the dosage loading was studied at 0.1–0.5 g and the result revealed that the removal efficiency of boron increased with a rise in the concentration of nano-Fe_3_O_4_.

#### 3.2.4. pH Dependency

The pH of the solution is greatly influenced by the surface charge, the amount of ionization, and the adsorbate speciation. Adsorption is considered to be low at acidic condition due to a high concentration of H3O+ that competes with the positively charged ions for the active binding site on the adsorbent surface, and this usually results in low contaminant uptake [30]. The effect of pH on the adsorption of boron on nano-magnetite was evaluated between the pH range of 4–9. Figure 11 demonstrates the removal efficiency of magnetite with regards to pH in the boron solution. 

An increase in the solution pH results in enhanced adsorption since ion exchange is more effective when fewer protons are available to compete with the boron ion for binding sites on the adsorbent surface. In aqueous solution, boron exists mostly as boric acid, which is predominantly weak. Furthermore, boron is attracted to the surface of the adsorbent to form a monolayer/multilayer surface. The best conditions for boron removal occur when there is no charge on the surface, and boric acid in solution is the predominant species. 

As shown in Figure 11, the results revealed that there is higher removal efficiency at pH 8. This result is in agreement with a previous study [63]. The surface sites are protonated below pH 8, boric acid dissociation on the surface is unlikely to occur, and therefore the efficiency of boron removal declines. At pH 9, a lower removal was recorded since the media were more negatively charged, thereby leading to repulsion between the nano-magnetite and borate ion [67]. This finding is in line with a previous study [3] where boron removal was achieved at higher pH. 

### 3.3. Regeneration Study

The re-usability test was undertaken in five adsorption/desorption cycles. Five phases of adsorption/desorption were conducted to evaluate the reusability of the synthesized nano-magnetite according to a previously adopted procedure [68]. The result of the present study showed a decrease in adsorptive efficiency after five cycles (Figure 12). Overall, the removal efficiency decreased from 75% to 69% after the first cycles for HCl solution desorption, showing less than 10% decline in efficiency, thereby demonstrating good performance and its capability for large-scale application. The recyclability experiment indicates that nano-magnetite can be used repeatedly in wastewater treatment as an effective adsorbent. The decrease in adsorption capacity of boron by nano-magnetite may be attributed to competitive adsorption and fewer available binding sites [69]. 

### 3.4. Boron Adsorption Isotherm

The experimental adsorption data obtained were evaluated for best fit with the Freundlich and Langmuir isotherm models to analyze the interrelationship between the adsorbent and the adsorbate. The fitting of the experimental data into the isotherm models describes the process of adsorption by the constants and correlation coefficient R^2^. To achieve this, log q_e_ is plotted against log Ce built on the linear form of the Freundlich model [70]. In Langmuir model linear expression, Ce/q_e_ is plotted against C_e_ (Figure 13) [71,72]. As shown in Figure 13, the initial concentration correlation coefficient with an R^2^ value of 0.9464 proves the Freundlich adsorption isotherm to be more favorable in the determination of the effectiveness of magnetite for the removal of boron. A summary of the isotherm parameters is presented in Table 1.

Freundlich constant K_F_ and 1/n were determined from the slope and intercept of the graphs (Figure 14). The Freundlich isotherm model adopts heterogeneous surface adsorption and explains the sorption onto the solid interface from a liquid medium [73]. The basic Freundlich constant that is usually employed to define the nature of the sorption process includes n and K_F_ and is calculated from the slope and intercept (Figure 14). Basically, the constant n represents the heterogeneity of the adsorption process at a magnitude of less than 1 (n < 1), equal to 1 (n = 1), or greater than 1 (n > 1) indicating the prevalence of linear chemical or physical adsorption [70]. Furthermore, 1/n constant reciprocal implies natural sorption; thus, the adsorption process is beneficial. The values of n, 1/n, K_F_, and R^2^ for the current work are presented in Table 1. 

It can be observed from Table 2 that the adsorption capacity of magnetite in the present study significantly exceeds the amount adsorbed by the other adsorbents. Similarly, the result of Taylor and Demetriou [50] showed high adsorption capacity at pH 8; however, the reusability studies were not considered. The high adsorption capacity of the nano-magnetite used in the present study could be attributed to the small pore size, high surface area, superparamagnetism, and high adsorptive capacity. There are limited studies on the use of in-house synthesized nano-magnetite from mill scale waste product for boron adsorption. Moreover, as the adsorbent exhibited high regeneration efficiency, the resultant nano-magnetite could be reused multiple times.

## 4. Conclusions

The nano-magnetite was successfully synthesized from mill scale waste using the high energy ball milling technique (HEBM), characterized, and then applied to remove boron from aqueous solution. The resultant nano-magnetite exhibited an excellent adsorption performance. The highest removal efficiency of 84% was attained. At the initial concentration of 50 mg/L, boron adsorption capacity of 8.4 mg/g was recorded. A reusability efficiency of 47% was obtained even after five cycles of desorption and reapplications. The Freundlich adsorption isotherm fits best with a correlation coefficient (R^2^) of 0.9464. Based on these results, it can be concluded that this nano-magnetite is a promising adsorbent for boron removal.

## Figures and Tables

**Figure 1 ijerph-18-01400-f001:**
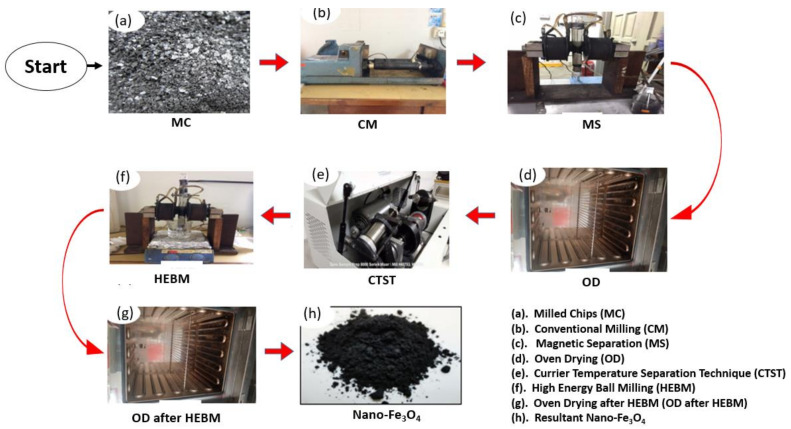
Procedure for the synthesis of magnetite nanoparticles (Fe_3_O_4_).

**Figure 2 ijerph-18-01400-f002:**
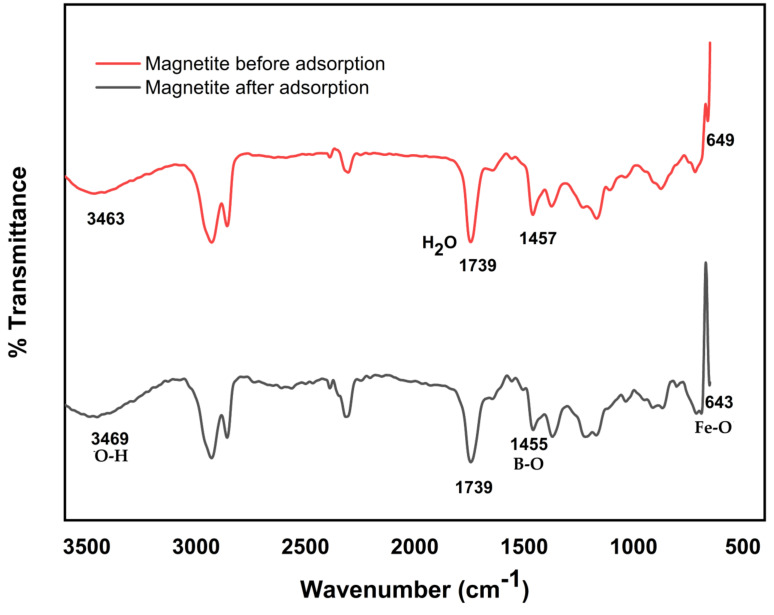
FTIR (Fourier-transform infrared spectroscopy) analysis of magnetite before and after adsorption of boron.

**Figure 3 ijerph-18-01400-f003:**
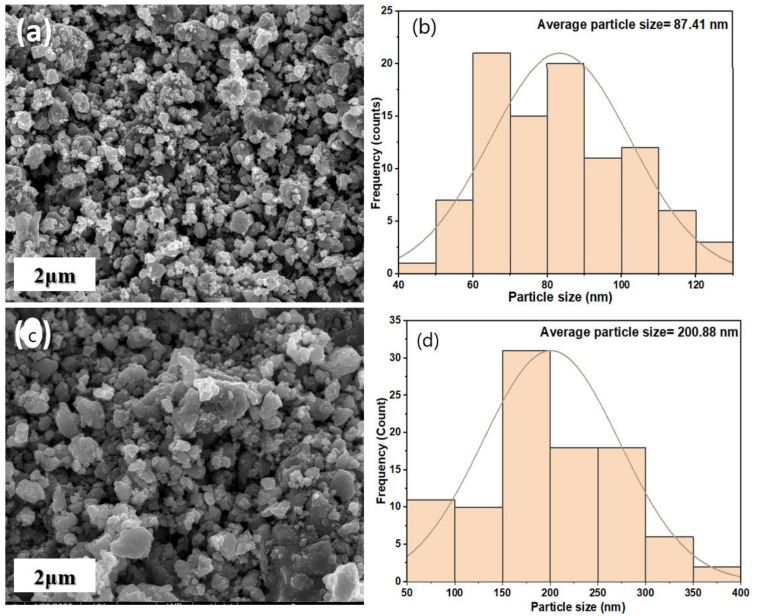
SEM (scanning electron microscopy) images for magnetite (**a**) before and (**c**) after adsorption; average particle size (**b**) before adsorption and (**d**) after adsorption.

**Figure 4 ijerph-18-01400-f004:**
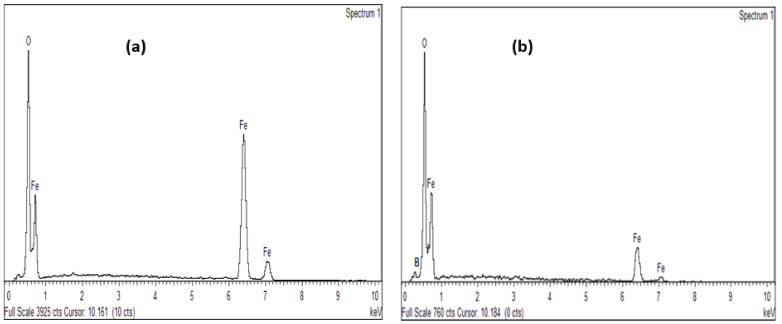
EDX spectra of the synthesized nano-magnetite (Fe_3_O_4_), (**a**) before adsorption and (**b**) after adsorption.

**Figure 5 ijerph-18-01400-f005:**
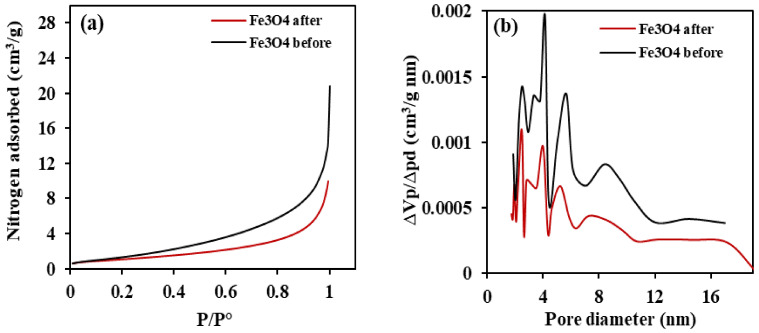
Graphical representation of (**a**) nitrogen adsorption isotherms at 77 K and nonlocal density functional theory (NLDFT) method of (**b**) micropore size distribution of magnetite.

**Figure 6 ijerph-18-01400-f006:**
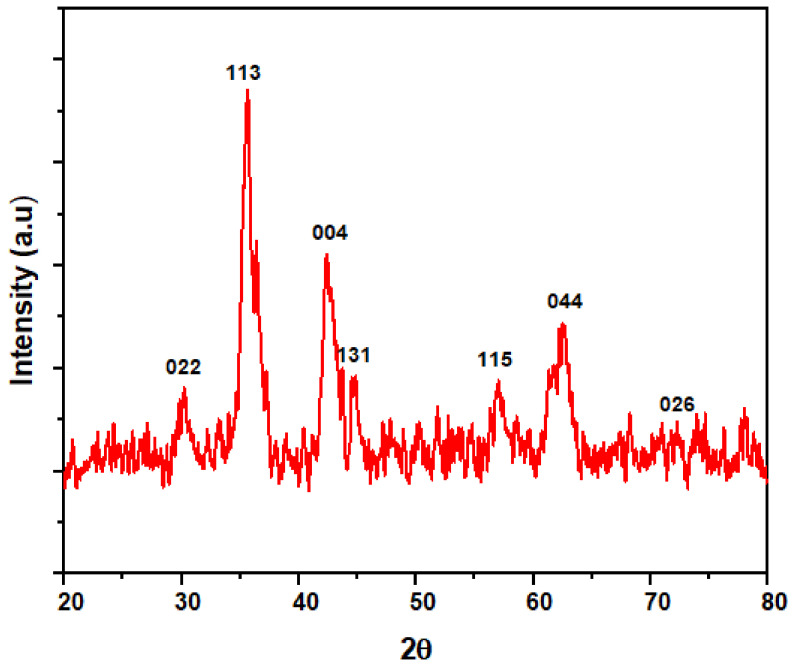
XRD (X-ray diffraction) spectra of the magnetic nanoparticles derived from mill scale waste.

**Figure 7 ijerph-18-01400-f007:**
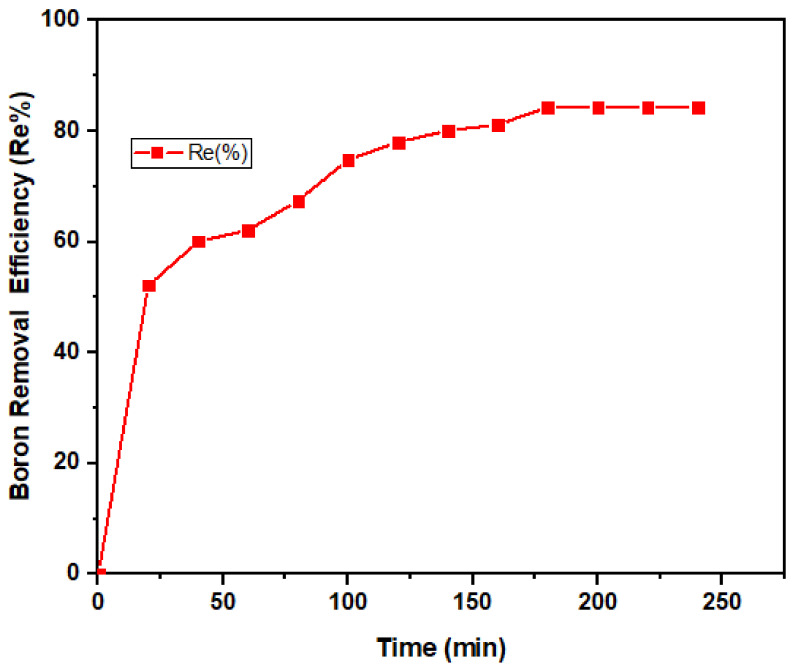
Boron removal efficiency by magnetite under variable contact times (20–240 min) at fixed dosage (0.5 g) and pH (8).

**Figure 8 ijerph-18-01400-f008:**
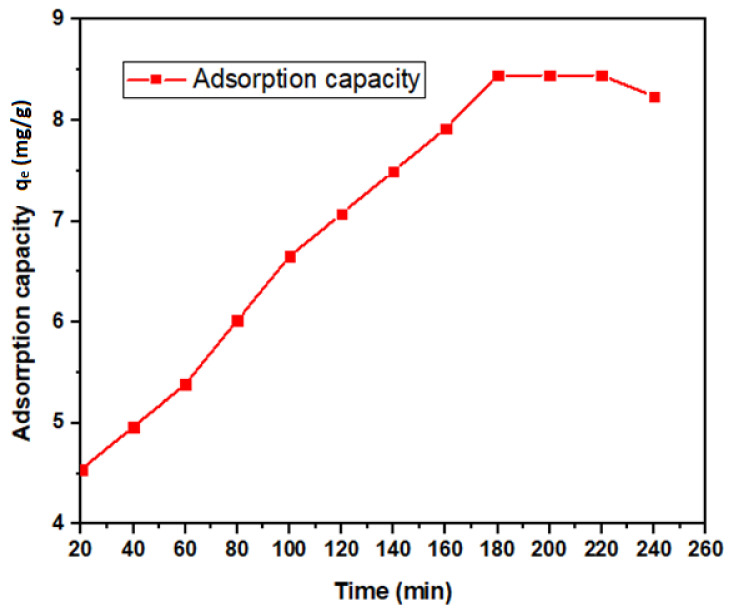
Boron adsorption capacity by nano-magnetite under variable contact times (20–240 min) at fixed dosage (0.5 g) and pH (8).

**Figure 9 ijerph-18-01400-f009:**
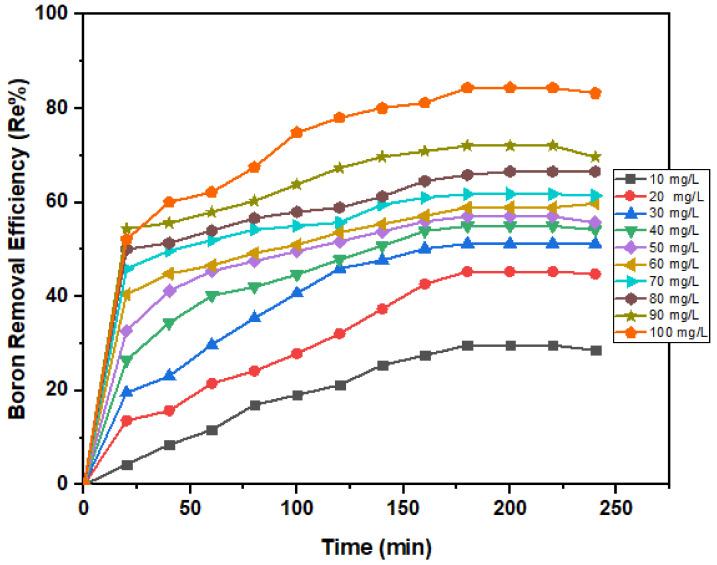
Boron removal efficiency by nano-magnetite under variable initial concentrations (10–100 mg/L) at constant pH (8), dosage (0.5 g), and variable contact times (20–240 min).

**Figure 10 ijerph-18-01400-f010:**
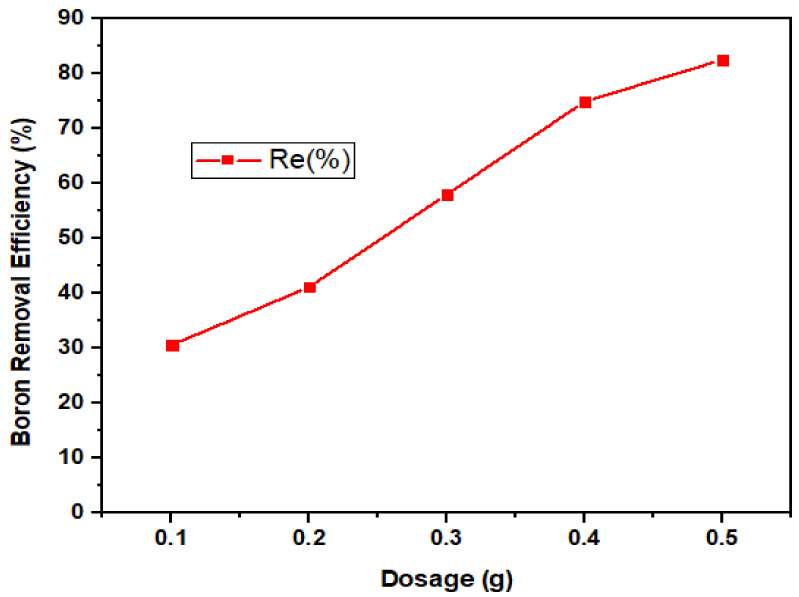
Boron removal efficiency by magnetite under variable dosages (0.1–0.5 g) at fixed pH (8) and contact time (180 min).

**Figure 11 ijerph-18-01400-f011:**
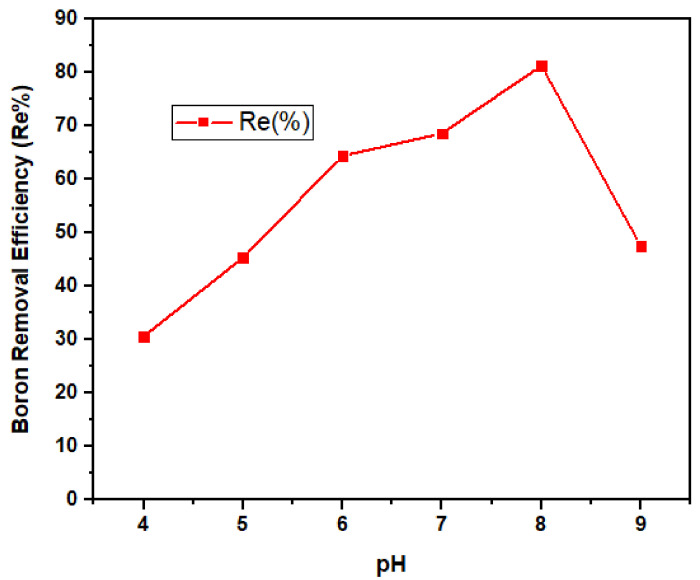
Boron removal efficiency by nano-magnetite under variable pH (4–9) at fixed dose (0.5 g) and contact time (180 min).

**Figure 12 ijerph-18-01400-f012:**
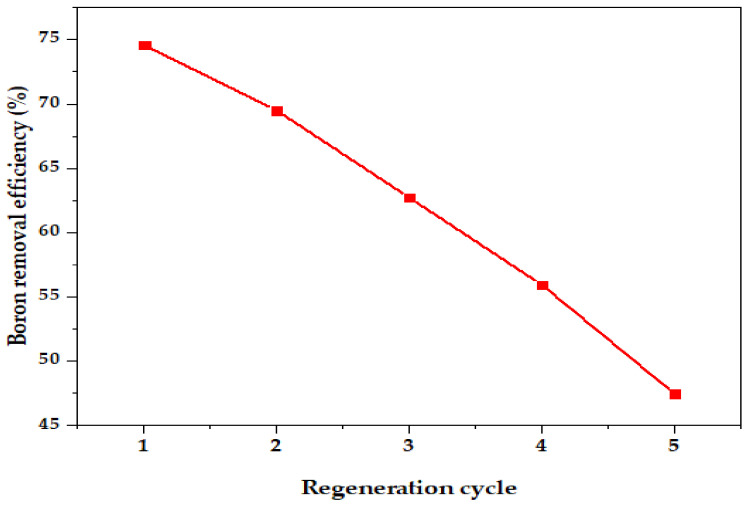
Reusability of nano-magnetite for boron adsorption.

**Figure 13 ijerph-18-01400-f013:**
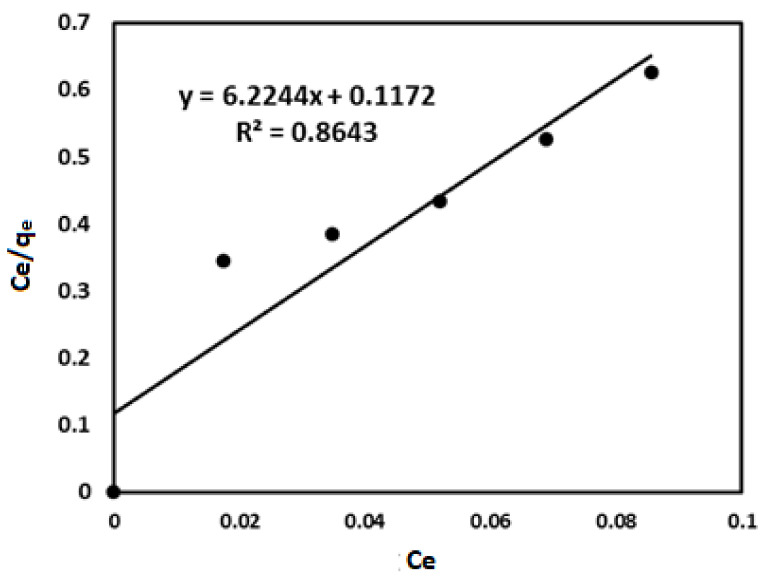
Langmuir isotherm on boron adsorption onto nano-magnetite at pH 8, dosage 0.5 g, contact time 240 min, and initial boron concentration 50 mg/L.

**Figure 14 ijerph-18-01400-f014:**
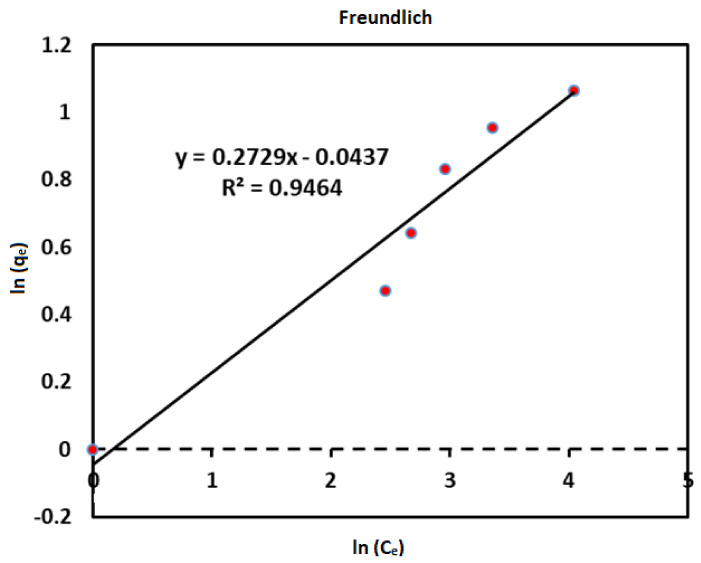
Freundlich isotherm on boron adsorption onto nano-magnetite at pH 8, dosage 0.5 g, contact time 240 min, and initial boron concentration 50 mg/L.

**Table 1 ijerph-18-01400-t001:** Isotherm constants for boron adsorption onto nano-magnetite.

Isotherm models	Parameters	Values
Freundlich	n	0.2729
	1/n	3.6643
	$K_{F}$ (mg/g)(l/mg)^1/n^	2.7183
	R^2^	0.9464
Langmuir	$q_{m}$ (mg/g)	0.1607
	$K_{L}$(L/mg)	0.7293
	R^2^	0.8643

**Table 2 ijerph-18-01400-t002:** Comparison of boron adsorption capacities for various adsorbents.

Adsorbents	pH	q_e_ (mg/g)	Re (%)	References
Magnetic nanoparticles improved with tartaric acid	6	1.97	ND	[73]
Magnetic magnetite (Fe_3_O_4_) nanoparticle	7	1.07	ND	[74]
Metallurgical slag and iron nanoparticles	8	NA	93.33	[75]
Boron-selective gel and commercial resin	8	1.15	ND	[56]
Gly-resin	9	1.60	96.90	[25]
Activated carbon	5.5	3.50	35	[68]
Iron oxide	8	6.94	ND	[50]
Cobalt (11)-doped chitosan bio-composite	8.5	2.50	ND	[26]
Activated alumina	8.5	---	65	[17]
Unmodified rice husk	5	4.23		[28]
Fe_3_O_4_	8	8.44	84	Current study

ND = Not determined. NA = Not available.

## Data Availability

The data in the present study is experimental data.

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
