# Peer review of "Synthesis of Nano-Magnetite from Industrial Mill Chips for the Application of Boron Removal: Characterization and Adsorption Efficacy"

_ijerph, 2021, doi:10.3390/ijerph18041400_

Round 1

Reviewer 1 Report

The authors described an inexpensive yet robust preparation of magnetite nanomaterial through industrial ball milling and evaluated the adsorption behaviors toward boron cations in aqueous solution. The characterization was extensive, including structural identification, surface morphologies, and a variety of adsorption conditions, as well as isotherm studies. In general, this manuscript is well-organized and presented in a logical manner. I recommend it for publication after addressing the following concerns:

1). The manuscript lacks quantitative data to show physical properties of the magnetite nanomaterial, especially, the size distributions.  What is the average size in nm? Is it evenly distributed? Without such data, it would be hard for others to reproduce the experiments if they use similar conditions but not quite the same.

2). The authors have not considered the interference of other elements. What is the selectivity of nano-magnetite adsorbent? If there’s Na+, Mg2+, Al3+, K+, in the testing environment, is it going to reduce the boron adsorption capacity and removal efficiency?

3). The authors measured the BET surface area and average pore size of the synthesized nano-magnetite. However, they didn’t provide such data in any graphic presentation. The authors mentioned that “Figure 4 presents before and after adsorption plots of nano-magnetite” (Page 7, Line 233). This is inconsistent with the actual Figure 4 in the manuscript, which is EDX characterization. The authors might have forgotten to include the appropriate graphs.

4). Several typos appeared in the manuscript, including but not limited to the following:

  1. i) “Magnetite before adsrption” in Figure 2.
  2. ii) “…by magnetite under variable dosage (0.5 - 0.5 g)” in Figure 9. The authors may want to double check the range.

5). Part of the body text occurred in between the graph and title of Figure 4 (Page 7, Line 224-226). This format issue needs to be corrected.

Author Response

Good day sir.

Kindly find the attached file containing the responses to reviewer 1

Reviewer 2 Report

Man et al. Describe the synthesis of nano-magnetite from mill scale waste using high energy ball milling technique. The compound obtained was fully characterized and used to remove boron from aqueous solutions, finding this compound to be a promising candidate for boron removal. In general, the chemistry presented seem to have been performed in a proper manner and the results are in line with the results presented. The manuscript is amenable for reading and certainly would be of interest for reading community of IJERPH. The references are fine in number and are actual. Based on the above the MS can be accepted in its present form.

Author Response

Good day sir.

Response to reviewer 2 attached.

Thank you

Reviewer 3 Report

1) Page 2 Line 61-68. Paragraph mentions that the adsorption technique presents far fewer challenges than membrane filtration or precipitation-coagulation. However, only few general challenges associated with boron removal have been mentioned in the manuscript .
Authors need to mention very specific advantages of the adsorption technique compared to membrane filtration and precipitation- coagulation. Each method has its own own advantages and disadvantages
2) There are several papers published on Boron removal using Fe3O4 nanoparticles. Need to mention the novelty of the current study compared to those published papers. Also need to cite those papers in this study. A few examples/papers are given below
a) Boron removal and reclamation by magnetic magnetite (Fe3O4) nanoparticle: An adsorption and isotopic separation study
b) Adsorption of Boron by Metallurgical Slag and Iron Nanoparticles
3) Figure 2 . Need common vertical curser passing through both spectra (on each major peaks). This could show the shift in peaks after adsorption. Without a common curser, it is hard to notice any shift in peaks after adsorption.
4) Resolution (quality) of SEM images are very poor. Need better high resolution images to characterize nano particle size.
5) Need to compare the boron adsorption capacity in the current study to previous studies using iron based nanoparticles (preferably in a table).
6) Need better conclusion. Avoid using quantitative results (values) in conclusion. They are already mentioned in abstract and results.
7) Minor Grammar revisions are required.

Author Response

Good day sir.

Find the attached response to reviewer 3.

Thank you
